# PARD: Accelerating LLM Inference with Low-Cost PARallel Draft Model Adaptation

**Zihao An**[1*]    **Huajun Bai**[1,2*]    **Ziqiong Liu**[1]    **Dong Li**[1]    **Emad Barsoum**[1]
[1]Advanced Micro Devices, Inc.    [2]Tsinghua University
{Zihao.An, Huajun.Bai, Ziqiong.Liu, d.li, Emad.Barsoum}@amd.com

## Abstract

The autoregressive nature of large language models (LLMs) fundamentally limits inference speed, as each forward pass generates only a single token and is often bottlenecked by memory bandwidth. Speculative decoding has emerged as a promising solution, adopting a draft-then-verify strategy to accelerate token generation. While the EAGLE series achieves strong acceleration, its requirement of training a separate draft head for each target model introduces substantial adaptation costs. In this work, we propose **PARD (PARallel Draft)**, a novel speculative decoding method featuring *target-independence* and *parallel token prediction*. Specifically, PARD enables a single draft model to be applied across an entire family of target models without requiring separate training for each variant, thereby minimizing adaptation costs. Meanwhile, PARD substantially accelerates inference by predicting multiple future tokens within a single forward pass of the draft phase. To further reduce the training adaptation cost of PARD, we propose a COnditional Drop-token (COD) mechanism based on the integrity of prefix key-value states, enabling autoregressive draft models to be adapted into parallel draft models at low-cost. Our experiments show that the proposed COD method improves draft model training efficiency by $3\times$ compared with traditional masked prediction training. On the `vLLM` inference framework, PARD achieves up to $3.67\times$ speedup on LLaMA3.1-8B, reaching **264.88** tokens per second, which is $1.15\times$ faster than EAGLE-3. Our code is available at `https://github.com/AMD-AGI/PARD`.

## 1 Introduction

Rapid advancements in LLMs such as GPT-4 (OpenAI, 2023), LLaMA3 (Llama Team, 2024) and DeepSeek-R1 (DeepSeek-AI et al., 2025) have fueled an explosion of applications such as content generation, code generation, and AI agents. However, as the counts of model parameters and the lengths of the context continue to grow, inference efficiency has become a critical challenge. Due to the auto-regressive (AR) nature of LLMs, tokens are generated sequentially, leading to substantial memory bandwidth consumption and high inference latency. Speculative Decoding (SD) (Chen et al., 2023a; Leviathan et al., 2023) has emerged as a promising technique to mitigate bandwidth overhead and reduce decoding latency during LLM inference. The core idea is to use a lightweight draft model to predict multiple candidate tokens, which are then verified in parallel by the target model. When combined with speculative sampling, this approach allows the model to generate multiple tokens within a single forward pass, significantly improving efficiency without compromising output quality.

The effectiveness of SD is determined jointly by the accuracy of the draft model and its overhead. To improve draft model accuracy, various methods have been proposed, primarily falling into the category of *target-dependent* approaches, where the draft model leverages information from the target model. For example, Medusa (Cai et al., 2024a) and EAGLE (Li et al., 2024a) incorporate features from the target model's outputs into the draft model's input, while LayerSkip (Elhoushi et al., 2024) and Kangaroo (Liu et al., 2024b) reuse selected layers of the target model as the draft model. Although these methods improve token prediction accuracy, they also introduce a major drawback: the draft model becomes *tightly coupled with the target model*. Consequently, any target model requires a separately trained draft model, which highly increases adaptation and deployment costs.

---
*Equal contribution.

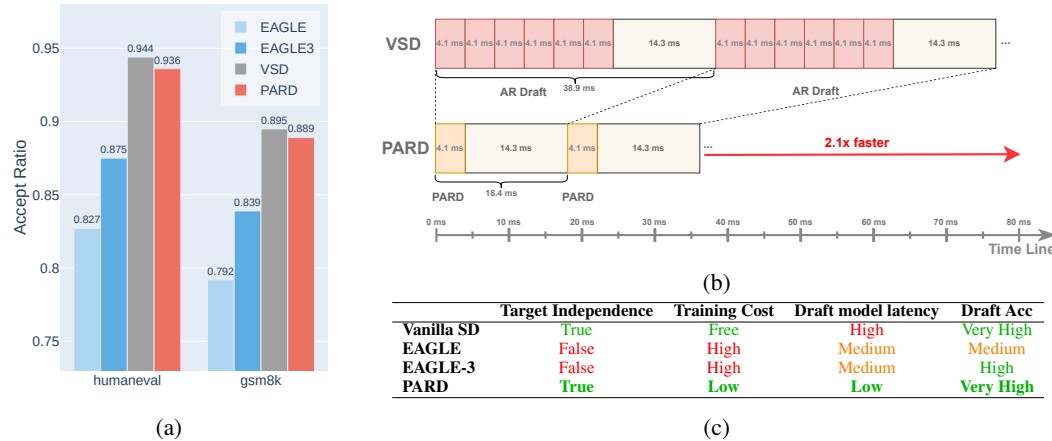

Figure 1: PARD achieves low latency while maintaining high accuracy. (a) Comparison of first-token acceptance rates using LLaMA3.1-8B as the target model. EAGLE and EAGLE-3 use their official model, vanilla speculative decoding (VSD) employs LLaMA3.2-1B as the draft model, and PARD represents the adapted version of VSD. (b) Comparison of actual inference time between VSD and PARD. VSD generates candidate tokens autoregressively during the draft stage, requiring multiple forward passes. In contrast, PARD completes drafting with a single forward pass. The draft model used is LLaMA3.2-1B and the target model is LLaMA3.1-8B. (c) Illustrative comparison of training and inference efficiency between PARD and other methods.

Unlike target-dependent methods, vanilla SD (Chen et al., 2023a; Leviathan et al., 2023) represents a class of *target-independent* approaches, where a single draft model can be applied across an entire family of target models without requiring separate training for each variant. For validation, we compare using LLaMA3.2-1B (Llama Team, 2024) as a draft model against the EAGLE method, both for the target model LLaMA3.1-8B. Figure 1a shows that LLaMA3.2-1B achieves significantly higher accuracy than the EAGLE head. However, due to its increased computational cost, the overall speedup ratio of vanilla SD may be lower than that of EAGLE.

To further accelerate inference, Mask-Predict (Ghazvininejad et al., 2019) provides a form of parallel decoding by using masked tokens as placeholders and employing specialized training to enable multiple token predictions in a single forward pass. Several recent studies have integrated SD with mask prediction. For example, PaSS (Monea et al., 2023) and BiTA (Gloeckle et al., 2024) fine-tune the target model into a parallel decoding model that functions as a draft model, while ParallelSpec (Xiao et al., 2024) extends EAGLE and Medusa into a parallel decoding framework. However, these approaches still use target model information, making them *inherently target-dependent*.

The goal of this paper is to break away from the target-dependent design paradigm, by developing a *target-independent* SD approach called **Parallel Draft (PARD)** with stronger generalization, higher inference efficiency, and lower adaptation cost. PARD builds upon existing high-accuracy small language models. A single draft model with low training cost can be applied across *an entire family of target models*, thereby significantly reducing adaptation overhead. PARD introduces mask tokens for parallel token predictions in the draft phase to accelerate inference, as shown in Figure 1b. To further improve training efficiency, we design a conditional drop-token (COD) mechanism based on the integrity of prefix key-value states, reducing the cost of adapting autoregressive draft models into parallel draft models by a factor of 3. The advantages of PARD are shown in Figure 1c.

To align our evaluation with real-world scenarios, all experiments are conducted on the widely adopted industrial inference framework vLLM. We also find that the inference performance of the Transformers library is not highly efficient, and thus apply corresponding software optimizations. Extensive experiments on LLaMA3 and Qwen familys show the consistent speedup of PARD over vanilla SD and EAGLE-3, as shown in Figure 2.

To summarize, we make the following contributions:

- We propose PARD, a novel speculative decoding method featuring *target-independence* and *parallel token prediction*. PARD is highly generalizable: its target-independent design allows

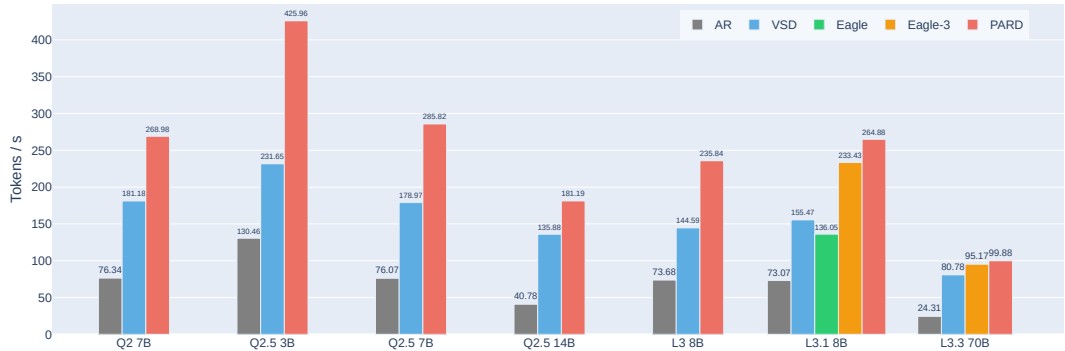

Figure 2: Performance comparison of different methods on the HumanEval task under vLLM. AR denotes the auto-regressive baseline, and VSD denotes vanilla speculative decoding, where the draft models used are LLaMA3.2-1B and Qwen2.5-0.5B.

a single draft model to accelerate an entire family of target models, in contrast to target-dependent methods such as Medusa and EAGLE. This significantly reduces deployment and adaptation costs. Through mask tokens, PARD substantially accelerates inference by predicting multiple future tokens within a single forward pass of the draft phase.

- We propose a COnditional Drop-token (COD) strategy for the effient training of PARD. Leveraging the *integrity of prefix key-value states*, COD enables low-cost adaptation of autoregressive draft models into parallel ones, boosting training efficiency by up to $3\times$ while maintaining accuracy.

- We integrate PARD into the high-performance inference framework vLLM. On LLaMA3.1-8B, PARD achieves a $3.67\times$ speedup, reaching a state-of-the-art throughput of **264.88** tokens per second on an A100-40GB GPU, which is $1.72\times$ faster than vanilla SD and $1.15\times$ faster than EAGLE-3.

## 2 PRELIMINARIES

### 2.1 AUTO-REGRESSIVE NATURE OF LLMS

Modern LLMs are based on the GPT architecture (Radford et al., 2018). During the training phase, GPT models leverage highly efficient parallelization, allowing tokens within a sequence to be processed simultaneously. This parallelism enables GPUs to fully utilize computational resources by maximizing matrix multiplications and optimizing memory bandwidth usage, thereby improving training efficiency. Given an input sequence $X = (x_0, x_1, \ldots, x_{N-1})$ and its corresponding target sequence $Y = (x_1, x_2, \ldots, x_N)$, the training objective of GPT is to minimize the auto-regressive loss, which can be expressed as:

$$\mathcal{L} = -\sum_{t=1}^{N} \log P(x_t | x_0, \ldots, x_{t-1}).$$ (1)

However, during inference, due to its auto-regressive nature, GPT must generate tokens sequentially, with each token depending on all previously generated tokens. This sequential generation process leads to a significant drop in computational efficiency. At each step $t$, the model computes:

$$x_t = \arg\max \ P(x | x_0, \ldots, x_{t-1}).$$ (2)

This sequential generation results in a memory-bound scenario, where GPU performance is constrained by the repeated loading of model weights and the KV-cache, rather than being fully utilized for large-scale parallel computations. As a result, even though GPUs are capable of high-speed matrix multiplications, the decoding phase remains bottlenecked by memory access latency, leading to high latency during the decoding phase.

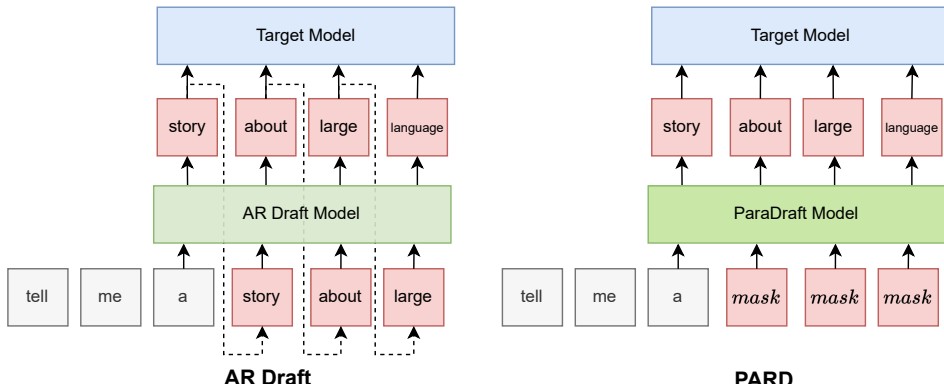

Figure 3: Illustration of PARD Inference. *Left:* Vanilla speculative decoding involves a draft model auto-regressively generating $K$ candidate tokens, which are then validated by the target model in parallel. *Right:* PARD introduces mask tokens for parallel Draft. All $K$ candidate tokens are generated in one forward pass.

## 2.2 SPECULATIVE DECODING

SD can effectively reduce decoding latency and improve matrix multiplication efficiency. The core idea behind SD is to first use a smaller draft model to generate a set of candidate tokens, and then use the target model to verify the candidates. The process is as follows:

**Drafting Stage:** The goal is to generate the next $K$ candidate tokens $C = (x'_n, \ldots, x'_{n+K-1})$. These candidates can be produced by a lightweight model or traditional machine learning algorithms. For the vanilla SD method, the draft model generates the $K$ candidates auto-regressively.

**Verification Stage:** The target model then verifies the candidate tokens in parallel, improving computational efficiency. When combined with speculative sampling, SD ensures no loss of performance.

We define $T_D$ as the time taken by the draft model for a single forward pass, and $T_T$ as the time taken by the target model for a single forward pass. The input length to both the draft and target models can vary across different SD methods. However, when the input length is not significantly large, the change in speed is negligible. Therefore, the time taken per iteration for predicting $K$ tokens by vanilla SD is:

$$T_{\text{AR}_{\text{draft}}} = K * T_D + T_T. \tag{3}$$

As shown in Figure 1b, when using a high acceptance rate model like LLaMA3.2-1B to accelerate LLaMA3.2-8B, the draft model consumes a considerable amount of time. Our PARD method fine-tunes the draft model into a parallel decoding model. In this case, the time taken is:

$$T_{\text{PARD}} = T_D + T_T. \tag{4}$$

It can be seen that PARD reduces the total time of the draft model to $1/K$ of the original time, thus significantly reducing the overall decoding time.

## 3 PARD FRAMEWORK

We begin by introducing how the PARD model predicts multiple tokens in a single forward pass and provide a detailed explanation of its inference process in Section 3.1. Next in Section 3.2, we describe how to finetune a vanilla draft model to acquire the capabilities of PARD while maitaining its target-independent feature. Moreover, we propose a conditional drop training token method that significantly accelerates training.

## 3.1 PARD INFERENCE

We incorporate the inference method Mask-Predict (Ghazvininejad et al., 2019) into PARD. The objective of the draft model $\theta$ is to predict the next $K$ tokens by optimizing the probability distribution:

$$P(x_n, \ldots, x_{n+K-1} | x_0, \ldots, x_{n-1}; \theta). \tag{5}$$

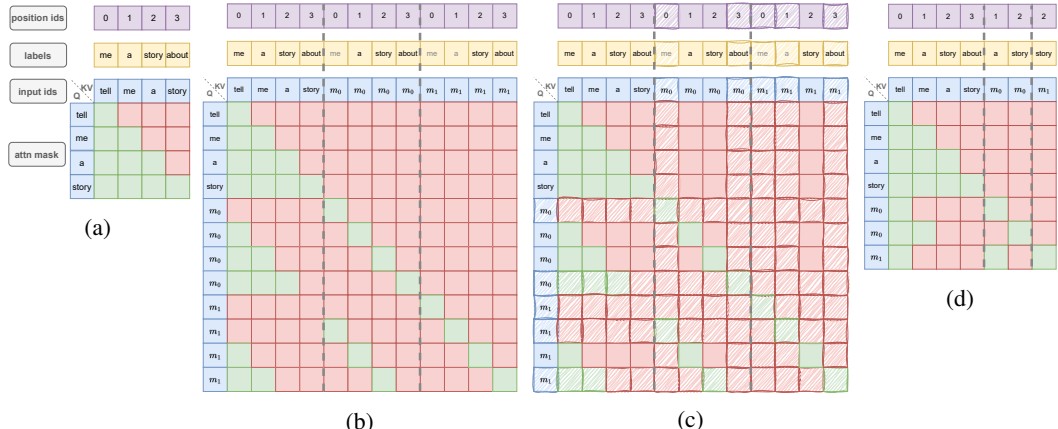

Figure 4: Illustration of Conditional Drop in PARD training. (a) Training data of the standard AR model. (b) Training data of PARD. The diagram is divided into three sections by dashed lines, corresponding to training objectives for predicting tokens at positions $+1$, $+2$, and $+3$. The designed attention mask ensures consistency between training and inference. Labels in lighter font indicate tokens that are supplemented for context completion and do not contribute to the loss computation. (c) Sparse training data for PARD with Conditional Drop, where shaded areas represent dropped tokens. The retention pattern follows a geometric decay with a fraction $r = 0.5$ of positions retained for mask token $m_0$ and $r^2 = 0.25$ for $m_1$, ensuring that each retained token maintains complete preceding key–value pairs. (d) The sparse matrix reorganized into a compact format by eliminating dropped positions.

For the vanilla SD method, this probability can be factorized as:

$$P(x_n, \ldots, x_{n+K-1}|x_0, \ldots, x_{n-1}; \boldsymbol{\theta}_{\mathrm{AR_{draft}}}) = \prod_{k=0}^{K-1} P(x_{n+k}|x_0, \ldots, x_{n+k-1}; \boldsymbol{\theta}_{\mathrm{AR_{draft}}}). \quad (6)$$

This formulation follows the standard autoregressive approach in Figure 3, where each token is generated sequentially based on all previously predicted tokens.

In contrast, PARD introduces a special token $m_k$ as a placeholder to replace tokens that would otherwise create dependencies. The formulation is as follows:

$$\begin{aligned} &P(x_n, \ldots, x_{n+K-1} \mid x_0, \ldots, x_{n-1}, \boldsymbol{\theta}_{\mathrm{PARD}}) \\ &= P(x_n \mid x_0, \ldots, x_{n-1}; \boldsymbol{\theta}_{\mathrm{PARD}}) \cdot \prod_{k=1}^{K-1} P(x_{n+k} \mid x_0, \ldots, x_{n-1}, m_0, \ldots, m_{k-1}; \boldsymbol{\theta}_{\mathrm{PARD}}). \end{aligned} \quad (7)$$

As shown in the equation, the predictions at each step do not depend on each other, enabling fully parallel inference. This reduces the number of forward passes required by the draft model from $K$ to 1, significantly improving inference efficiency.

## 3.2 PARD TRAINING

To enable a vanilla draft model to predict multiple tokens within a single forward pass, we adopt a mask-token training strategy, which adapts a standard autoregressive draft model into a parallel draft model. Unlike conventional approaches, we further propose a conditional drop-token (COD) mechanism based on the integrity of prefix key-value states, which significantly improves training efficiency while maintaining final performance.

### 3.2.1 MASK TOKENS BASED TRAINING

To ensure consistency between the training and inference processes, we divide training into multiple independent subtasks, where each subtask is designed to predict $x_n, \ldots, x_{n+K-1}$. As shown in Figure 4a 4b, these subtasks can be trained simultaneously with minimal preprocessing of the training

data. The loss for each subtask is computed using a cross-entropy function, as follows:

$$
\mathcal{L}_k = \begin{cases} -\dfrac{1}{N} \sum_{i=1}^{N} \log P(x_i \mid x_0, x_1, \ldots, x_{i-1}; \boldsymbol{\theta}_{\mathrm{PARD}}), & k = 1, \\ -\dfrac{1}{N-k+1} \sum_{i=k}^{N} \log P(x_i \mid x_0, x_1, \ldots, x_{i-k}, m_0, \ldots, m_{k-2}; \boldsymbol{\theta}_{\mathrm{PARD}}), & k > 1. \end{cases}
\tag{8}
$$

where $\mathcal{L}_k$ represents the loss function for predicting the $k$-th next token, and $N$ denotes the length of the sample.

### 3.2.2 CONDITIONAL DROP OF TOKENS

Compared to standard auto-regressive model training, the mask token training method breaks the task into multiple subtasks, significantly increasing the training cost. For example, if the training sample length is $N$, then the number of tokens for AR training is also $N$. However, in the mask token training approach, the number of tokens for training increases from $N$ to $K \times N$, where $K$ is the number of tokens the draft model predicts simultaneously in a single forward pass.

To address this challenge, we propose an innovative COnditional Drop token (COD) strategy. This method boosts training efficiency significantly while maintaining prediction accuracy. The core idea behind COD is that tokens in earlier subtasks are more critical, whereas those in later subtasks can be selectively dropped to reduce computation.

During the token dropping process, randomly dropped tokens can result in incomplete key and value states during attention computation. The COD mechanism ensures that the remaining tokens still provide complete key and value information when computing attention, thereby preserving essential contextual representations. Figure 4c illustrates the token dropping process. Although some tokens are removed, the key contextual information remains intact. Figure 4d shows the final reorganized data after token dropping.

To manage the number of tokens retained for each subtask, we introduce a retention parameter $r$. For the $i$-th subtask, the number of retained tokens $N_i$ is given by:

$$
N_i = N * r^{i-1}.
\tag{9}
$$

Consequently, the total number of training tokens can be expressed as:

$$
N_{\mathrm{COD}} = \sum_{i=1}^{K} N_i = \sum_{i=1}^{K} N * r^{i-1} = N \frac{1 - r^K}{1 - r} < \frac{N}{1 - r}.
\tag{10}
$$

For instance, when $r = 0.5$, the number of training tokens can be reduced from $K \times N$ to $2N$, significantly lowering the training cost. Additionally, to prevent excessive token dropping in later subtasks, we introduce a minimum retention ratio $r_{min}$, ensuring that the retention rate does not fall below a predefined threshold. The adjusted number of retained tokens for each subtask is:

$$
N_i' = N * \max(r^{i-1}, r_{\min}).
\tag{11}
$$

The detailed algorithm of COD, along with the mechanism integrity of prefix key-value states is provided in Appendix B. By employing the COD strategy, the training cost of PARD can be significantly reduced from $\mathcal{O}(N \cdot K)$ to $\mathcal{O}(N)$.

### 3.2.3 TRAINING EFFICIENCY

We evaluate the training cost of different speculative decoding methods in Pflops per 1M tokens, using LLaMA3.3-70B as the target model. PARD achieves a training efficiency that is $\mathbf{7}\times$ higher than EAGLE and $\mathbf{10}\times$ higher than EAGLE-3. Full derivations of forward, backward, and total training costs for each method are provided in Appendix A.

In addition to efficiency, PARD exhibits *target independence*, allowing a single draft model to accelerate an entire series of target models (e.g., LLaMA3-8B, 70B, 405B). In contrast, EAGLE-style methods require a separate draft model for each target, which significantly increases adaptation cost.

Table 1: Performance comparison of different methods on the Qwen and LLaMA3 series under the `vLLM` framework. The EAGLE series uses the official model.

| Target | Method | HumanEval | | GSM8K | | SpecBench | | Average | |
|---|---|---|---|---|---|---|---|---|---|
| | | TPS | SpeedUp | TPS | SpeedUp | TPS | SpeedUp | TPS | SpeedUp |
| Q2 7B | AR | 76.34 | 1.00 | 76.12 | 1.00 | 75.97 | 1.00 | 76.14 | 1.00 |
| | VSD | 181.18 | 2.37 | 201.15 | 2.64 | 112.00 | 1.47 | 164.78 | 2.16 |
| | PARD | **268.98** | **3.52** | **221.43** | **2.91** | **135.05** | **1.78** | **208.49** | **2.74** |
| Q2.5 3B | AR | 130.46 | 1.00 | 130.42 | 1.00 | 129.95 | 1.00 | 130.28 | 1.00 |
| | VSD | 231.65 | 1.78 | 238.71 | 1.83 | 136.26 | 1.05 | 202.21 | 1.55 |
| | PARD | **425.96** | **3.27** | **406.17** | **3.11** | **203.05** | **1.56** | **345.06** | **2.65** |
| Q2.5 7B | AR | 76.07 | 1.00 | 75.93 | 1.00 | 75.97 | 1.00 | 75.99 | 1.00 |
| | VSD | 178.97 | 2.35 | 162.85 | 2.14 | 107.38 | 1.41 | 149.73 | 1.97 |
| | PARD | **285.82** | **3.76** | **292.56** | **3.85** | **145.64** | **1.92** | **241.34** | **3.18** |
| Q2.5 14B | AR | 40.78 | 1.00 | 40.92 | 1.00 | 40.78 | 1.00 | 40.83 | 1.00 |
| | VSD | 135.88 | 3.33 | 146.69 | 3.58 | 80.59 | 1.98 | 121.05 | 2.97 |
| | PARD | **181.19** | **4.44** | **182.25** | **4.45** | **91.33** | **2.24** | **151.59** | **3.71** |
| L3 8B | AR | 73.68 | 1.00 | 73.57 | 1.00 | 72.80 | 1.00 | 73.35 | 1.00 |
| | VSD | 144.59 | 1.96 | 123.76 | 1.68 | 102.76 | 1.41 | 123.70 | 1.69 |
| | PARD | **235.84** | **3.20** | **200.45** | **2.72** | **147.80** | **2.03** | **194.70** | **2.65** |
| L3.1 8B | AR | 73.07 | 1.00 | 73.38 | 1.00 | 72.70 | 1.00 | 73.05 | 1.00 |
| | VSD | 155.47 | 2.13 | 140.93 | 1.92 | 106.44 | 1.46 | 134.28 | 1.84 |
| | EAGLE | 136.05 | 1.86 | 110.81 | 1.51 | 100.11 | 1.38 | 115.66 | 1.58 |
| | EAGLE-3 | 233.43 | 3.19 | 192.56 | 2.62 | 155.58 | 2.14 | 193.86 | 2.65 |
| | PARD | **264.88** | **3.63** | **235.09** | **3.20** | **157.49** | **2.17** | **219.15** | **3.00** |
| L3.3 70B | AR | 24.31 | 1.00 | 24.27 | 1.00 | 23.95 | 1.00 | 24.18 | 1.00 |
| | VSD | 80.78 | 3.32 | 77.07 | 3.18 | 53.09 | 2.22 | 70.31 | 2.91 |
| | EAGLE-3 | 95.17 | 3.91 | 75.34 | 3.10 | **63.14** | **2.64** | 77.88 | 3.22 |
| | PARD | **99.88** | **4.11** | **95.58** | **3.94** | 57.41 | 2.40 | **84.29** | **3.49** |

# 4 EVALUATION

## 4.1 EXPERIMENTAL SETUP

**Models**: We conduct experiments on popular industry models, including LLaMA3 (Grattafiori et al., 2024), DeepSeek-R1-Qwen (Guo et al., 2025), and Qwen (Yang et al., 2024). For each model series, we select the smallest model variant and train it in the PARD framework.

**Datasets**: To ensure alignment with the original instruct model training process, we select datasets tailored to each model series. LLaMA3 is trained with Magpie-Llama-3.1-Pro-1M (Xu et al., 2024) and Evol-CodeAlpaca (Luo et al., 2023). Qwen2.5 is trained with Magpie-Qwen2-Pro-1M and Evol-CodeAlpaca. DeepSeekR1Qwen is trained with OpenR1-Math-220k (Face, 2025), OpenThoughts-114k (Team, 2025), and Chinese-DeepSeek-R1-Distill-Data-110k (Liu et al., 2025). For LLaMA 3 and Qwen, we further enhance training accuracy by regenerating answers.

**Tasks**: We evaluate the effectiveness of PARD on mathematical reasoning and code generation tasks on benchmarks including HumanEval (Chen et al., 2021), GSM8K (Cobbe et al., 2021), MATH500 (Lightman et al., 2023) and SpecBench (Xia et al., 2024b).

**Training**: We conduct training on 8xMI250X cards using the TRL framework for a total of 4 epochs. During training, the parameters are set as follows: $k = 8$, $r = 0.7$, and $r_{min} = 0.2$. The detailed training hyperparameters are provided in Appendix C.

**Evaluation**: To better reflect real-world usage scenarios, all comparative experiments are conducted on the high-performance `vLLM` framework, and the evaluation is performed on A100-40GB GPU.

**Metrics**: Tokens Per Second: The number of tokens generated per second in real-world scenarios. Speedup: The acceleration ratio compared to the baseline standard auto-regressive generation method.

## 4.2 EXPERIMENTAL RESULTS

Table 1 compares the acceleration effects of PARD on the Qwen and LLaMA3 series, while Appendix E further reports results on the DeepSeek-Qwen series. On code tasks, PARD achieves

Table 2: Comparison of acceptance rates for PARD and EAGLE on LLaMA3.1-8B, where $k$-$\alpha$ denotes the average acceptance rate when the draft length is $k$.

| Method | HumanEval | | GSM8K | |
|---|---|---|---|---|
| | 1-$\alpha$ | 4-$\alpha$ | 1-$\alpha$ | 4-$\alpha$ |
| EAGLE | 0.83 | 0.72 | 0.79 | 0.66 |
| EAGLE-3 | 0.87 | 0.85 | 0.82 | 0.79 |
| PARD | **0.93** | **0.90** | **0.88** | **0.85** |

Table 3: Memory bandwidth usage during the draft phase of LLaMA3.1-8B model in bf16 dtype. PARD bandwidth usage remains constant as $k$ increases.

| Method | $k = 4$ | $k = 6$ | $k = 8$ |
|---|---|---|---|
| | Draft BW Consumption | | |
| EAGLE | 5.94 GB | 8.90 GB | 11.88 GB |
| EAGLE-3 | 5.94 GB | 8.90 GB | 11.88 GB |
| PARD | 2.48 GB | 2.48 GB | 2.48 GB |

speedups ranging from 3.20× to 4.44×, with average speedups between 2.65× and 3.71×. Notably, LLaMA3.1-8B runs 1.9× faster than EAGLE and 1.15× faster than EAGLE-3.

Our experiments further demonstrate the **target-independence** property of PARD, where a single PARD model can accelerate an entire series of target models, as shown in Table 1 and Figure 2. Specifically, we evaluate three target models from the LLaMA3 series and three target models from Qwen. In contrast, target-dependent methods such as the EAGLE series require separate training for each individual model. PARD achieves high acceleration without the need for model-specific adaptation, substantially lowering the barrier for deployment.

During our experiments, we observed that inference throughput using the Transformers library is significantly lower than that of vLLM Appendix F. For methods such as vanilla SPD, this suboptimal performance of Transformers can lead to relative speedups under the same inference framework being lower than those measured on vLLM (e.g., 1.36× and 2.26×). To better reflect real-world usage scenarios, all final experiments in this paper are conducted on the vLLM framework.

## 4.3 ABLATION STUDIES

For all ablation study experiments, the target model used is DeepSeek-R1-Qwen-7B, and the draft model's pretraining model is DeepSeek-R1-Qwen-1.5B. Training is conducted on a 93K subset of OpenR1-Math-220K for one epoch, and testing is performed using the MATH500 dataset.

**Conditional Drop Token**: The lower the retention rate, the greater the acceleration effect. However, excessively low retention may degrade model performance. As shown in 5a, when setting $r = 0.7$ and $r_{\min} = 0.2$, we achieve a good balance between speed and accuracy. This setting allows us to achieve 3× faster training while maintaining the original accuracy. All experiments in this paper adopt these parameters.

**Shared Mask Token ID Strategy**: We compare different prediction strategies and find that using the same mask token ID across all predicted positions, i.e., $m_0 = m_1 = \cdots = m_{K-1}$, performs better than using distinct token IDs. The corresponding throughput results are 221.97 and 218.05 tokens/s, respectively. This approach not only improves prediction consistency but also enhances the model's ability to generalize beyond its training configuration, a property we refer to as *extrapolation capability*. Specifically, extrapolation capability allows the model to infer with a larger $K$ during inference than it was trained on.

**Selection of Draft $K$**: We conduct a Cartesian product test for $K_{\text{train}}$ during training and $K_{\text{infer}}$ during inference in Figure 5b. Due to the extrapolation capability of PARD enabled by the shared mask token ID, $K_{\text{infer}}$ can be greater than $K_{\text{train}}$. The best performance is achieved at $K_{\text{infer}} = 12$, while results remain stable when $K_{\text{train}} \geq 8$. Therefore, we select $K_{\text{train}} = 8$.

**Comparison with the Mainstream Method EAGLE**: For SD, higher acceptance ratio combined with lower bandwidth consumption results in superior speedup. In Table 2 and Table 3, we compare PARD and EAGLE series. PARD achieves a higher acceptance ratio while consuming less bandwidth.

**Large Batch Size Inference**: Appendix D reports results with batch sizes ranging from 1 to 16. As the batch size increases, the bottleneck shifts from memory-bound to compute-bound. In this setting, PARD achieves speedups between 1.33× to 3.63×.

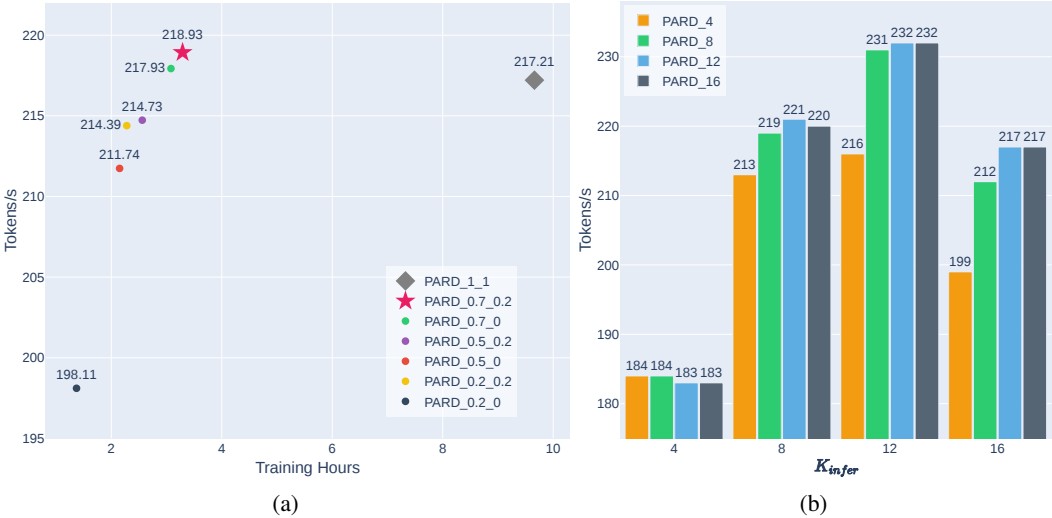

Figure 5: (a) Compare the effects of different values of $r$ and $r_{\min}$, where each experiment is labeled as PARD_$r$_$r_{\min}$. The x-axis represents training time, while the y-axis indicates the final decoding speed. (b) Present the results under different $K_{train}$ and $K_{infer}$ settings. The x-axis represents $K_{infer}$, and the experiment names PARD_$K_{train}$ denote different $K_{train}$ values.

## 5 RELATED WORK

Improving the inference efficiency of large language has been extensively studied from multiple perspectives. Quantization techniques such as GPTQ (Frantar et al., 2022), AWQ (Lin et al., 2024), SmoothQuant (Xiao et al., 2023a), and LLM-QAT (Liu et al., 2023) focus on reducing computational and memory costs. To address long-context KV-cache management, approaches such as GQA (Ainslie et al., 2023), MLA (Liu et al., 2024a), StreamingLLM (Xiao et al., 2023b), H2O (Zhang et al., 2023), MoBA (Lu et al., 2025), and NSA (Yuan et al., 2025) explicitly balances GPU memory consumption against model accuracy. At the system level, innovations like FlashAttention (Dao et al., 2022), FlashDecoding (Hong et al., 2023), MEGABLOCKS (Gale et al., 2023), and vLLM (Kwon et al., 2023) deliver optimized kernels and scheduling strategies to maximize hardware utilization and throughput.

Speculative decoding (Leviathan et al., 2023) (Chen et al., 2023b) improve GPU parallelism by leveraging a draft model to generate candidate tokens, which are then verified by the target model, achieving speedup without compromising accuracy. Other approaches such as LOOKAHEAD (Fu et al., 2024), PLD+ (Somasundaram et al., 2024), REST (He et al., 2023) and SuffixDecoding (Oliaro et al., 2024) utilize text-based retrieval mechanisms to generate more informed drafts. LayerSkip (Elhoushi et al., 2024), Kangaroo (Liu et al., 2024b), and SWIFT (Xia et al., 2024a) reuse selected layers of the target model to construct a lightweight draft model. CSDrafting (Chen et al., 2024b) employs a cascade speculative draft model. CLLMs (Kou et al., 2024) and NSN (Wang et al., 2024) adopt Jacobi decoding.

To improve the accuracy of speculative decoding, methods like Medusa (Cai et al., 2024b), EAGLE (Li et al., 2024b), EAGLE-3 (Li et al., 2025), Amphista (Li et al., 2024c) and Hydra (Ankner et al., 2024) incorporate representations from the target model as additional input signals. Approaches such as BiTA (Gloeckle et al., 2024), ParallelSpec (Xiao et al., 2024), and PaSS (Monea et al., 2023) introduce mask tokens to enable parallel speculative decoding. Further, techniques including Spectr (Sun et al., 2023), SpecInfer (Miao et al., 2023), Sequoia (Chen et al., 2024a), and EAGLE-2 (Li et al., 2024b) optimize tree-based verification structures to enhance token acceptance rates.

## 6 CONCLUSION

We presented PARD, a novel speculative decoding method that is *target-independent* and supports *parallel token prediction*. Unlike existing target-dependent approaches such as Medusa and EAGLE,

PARD allows a single draft model to accelerate an entire family of target models, significantly reducing adaptation and deployment costs. To improve training efficiency, we proposed the **Conditional Drop-token (COD)** mechanism, which leverages the integrity of prefix key-value states to adapt autoregressive draft models into parallel ones at a fraction of the cost. Extensive experiments on LLaMA3 and Qwen families show that PARD consistently outperforms vanilla speculative decoding and EAGLE-based methods. On LLaMA3.1-8B, PARD achieves **264.88 tokens per second**, corresponding to a **3.67×** speedup over standard autoregressive inference and **1.15×** speedup over EAGLE-3. In summary, PARD offers a highly generalizable, efficient, and practical framework for accelerating large language model inference, demonstrating the potential of target-independent and parallel decoding strategies for scalable LLM deployment.

## ETHICS STATEMENT

This work adheres to the ICLR Code of Ethics. It does not involve human subjects, sensitive data, or applications with foreseeable harmful impact. All datasets and methods are used in compliance with ethical and legal standards.

## REPRODUCIBILITY STATEMENT

The experimental hyperparameters are detailed in Appendix C. All reported results are reproducible, and both code and models will be released.

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

# A    TRAINING COST

**Comparison setting:** We measure training cost in Pflops per 1M tokens, with the target model fixed to LLaMA3-70B.

**PARD:** For non-long-text scenarios, the forward pass cost is approximated as twice the model size, where the factor 2 accounts for both multiply and add operations:

$$Pflops_{\text{PARD},F} \approx 2 \times \text{ParameterSize} \times \text{InputTokenNum}/10^{15} = 2 \times 10^9 \times 3.37 \times 10^6/10^{15} = 6.74,$$

where 3.37M is the effective input token count after applying mask prediction with COD. The backward pass requires twice the forward cost:

$$Pflops_{\text{PARD},B} \approx 2 \times 6.74 = 13.48,$$

giving a total training cost of

$$Pflops_{\text{PARD}} \approx 20.22.$$

**EAGLE:** The forward pass includes both the target model (70B) and the draft model (0.8B layer + 1.01B lm-head):

$$Pflops_{\text{EAGLE},F} \approx 2 \times (70 + 0.86 + 1.01) \times 10^9 \times 10^6/10^{15} = 143.74.$$

For the backward pass, the target model requires no gradient computation, the draft layer costs twice its forward, and the lm-head costs one forward (activation gradients only):

$$Pflops_{\text{EAGLE},B} \approx 2 \times \left(2 \times 0.86 + 1.01\right) \times 10^9 \times 10^6/10^{15} = 5.46,$$

yielding a total training cost of

$$Pflops_{\text{EAGLE}} \approx 149.20.$$

**EAGLE-3:** Compared with EAGLE, the training strategy uses HASS (Zhang et al., 2024), and the input tokens for the EAGLE-3 head are seven times that of EAGLE. For the forward pass:

$$Pflops_{\text{EAGLE3},F} \approx 2 \times \left(70 + 7 \times (0.86 + 1.01)\right) \times 10^9 \times 10^6/10^{15} = 166.18.$$

For the backward pass:

$$Pflops_{\text{EAGLE3},B} \approx 2 \times 7 \times \left(2 \times 0.86 + 1.01\right) \times 10^9 \times 10^6/10^{15} = 38.22,$$

yielding a total training cost of

$$Pflops_{\text{EAGLE3}} \approx 204.4.$$

For training draft models on LLaMA3-70B, PARD achieves a training efficiency that is $7\times$ higher than EAGLE and $10\times$ higher than EAGLE-3.

Target Independence: PARD's target independence allows a single draft model to serve an entire model series (e.g., LLaMA3-8B, 70B, 405B), while EAGLE-style methods require a separate draft for each target.

## B  Conditional Drop Tokens Algorithm

A naive random token drop could indeed break the completeness of key and value information during training. Algorithm 1 presents the pseudocode of the COD data processing procedure. Below we provide an illustrative example of how tokens are selected for dropping, based on Figure 4:

- For clarity, we further index each original token $m_i$ in the figure as $m_{i,j}$ for $j \in \{0,1,2,3\}$, where $j$ indicates the position within $m_i$.
- In Figure 4b, without COD, the training sequence is divided into three sections (delineated by dashed lines), corresponding to prediction objectives for positions $+1$, $+2$, and $+3$ using all tokens.
- In Figure 4c, with COD applied, shaded regions denote dropped tokens. We use a geometric-decay retention rate $r = 0.5$:
    - For position $+1$, no tokens are dropped in the leftmost section, so the full context [tell, me, a, story] is used.
    - For position $+2$, we drop $(1-r) = 50\%$ of tokens in the middle section. Specifically, we drop $m_{0,0}$ and $m_{0,3}$, and retain $m_{0,1}$ and $m_{0,2}$.
    - For position $+3$, we drop $(1-r^2) = 75\%$ of tokens in the rightmost section but ensure complete prefix key and value context for each retained token. For example, $m_{1,3}$ (predicting "about" from [tell, me, $m_{0,2}$]) can be retained because all its prefix tokens remain. Similarly, $m_{1,2}$ can be retained since its prefixes [tell, $m_{0,1}$] are intact, whereas $m_{1,1}$ is excluded because its prefix $m_{0,0}$ was dropped at position $+2$. From the valid candidates $\{m_{1,2}, m_{1,3}\}$, we randomly select one token (here $m_{1,2}$).
- In Figure 4d, new data after COD.

Regarding COD's effect, we conducted an ablation study in Figure 5a. The results show that COD can speed up training by threefold while maintaining the same inference acceleration on the target model.

---

**Algorithm 1** PARD with Conditional Drop Tokens: Data Processing

---

1: **Input:** Training dataset $\mathcal{D}$, PARD prediction count $K$, retention decay factor $r$, minimum retention rate $r_{\min}$
2: **Output:** Processed training data with updated input_ids, labels, position encodings, and attention masks
3: **for** each data sample $X \in \mathcal{D}$ **do**
4:     $X \leftarrow [X_1, \dots, X_K]$, where $X_k$ is the training data for predicting the $k$-th token, including input ids, label, position ids, and attention mask
5:     **for** $k = 1$ to $K$ **do**
6:         Compute retention rate: $\gamma \leftarrow \max(r^{k-1}, r_{\min})$
7:         Decide which tokens in $X_k$ to retain, ensuring that the preceding KV cache for attention computation is complete
8:         Update $X_k$ (i.e., input ids, label, position ids, and attention mask) to obtain $X'_k$
9:     **end for**
10:     Merge updated sequences $X' \leftarrow [X'_1, \dots, X'_K]$ and update the overall attention mask
11:     Store the processed data for this sample
12: **end for**

---

## C  Training Hyperparameters

Table 4 summarizes the hyperparameters used for training.

## D  Large Batch Size Inference

Table 5 presents results across batch sizes ranging from 1 to 16. As the batch size increases, the bottleneck shifts from memory-bound to compute-bound. Under these conditions, PARD achieves a speedup of $1.33\times$ to $3.63\times$.

Table 4: Selected Hyperparameters for PARD Training

| Hyperparameter | Llama3 | Deepseek-R1-Qwen | Qwen |
|---|---|---|---|
| Optimizers | AdamW | AdamW | AdamW |
| Learning Rate | 1e-5 | 3e-5 | 8e-5 |
| Per Device Train Batch Size | 4 | 4 | 8 |
| Gradient Accumulation Steps | 2 | 2 | 1 |
| Num Processes | 8 | 8 | 8 |
| Num Train Epochs | 4 | 4 | 4 |
| Training PARD K | 8 | 8 | 8 |
| Max Seq Length | 512 | 1024 | 512 |
| PARD Token ID | 128020 | 151665 | 151665 |
| The Answer Of Training Data | Regenerate + Original | Original | Regenerate |

Table 5: Performance comparison across different batch sizes on LLaMA3.1-8B in the vLLM framework, evaluated on HumanEval.

| Method | bs=1 | bs=2 | bs=4 | bs=8 | bs=16 |
|---|---|---|---|---|---|
| | Speedup | | | | |
| AR | 1.00 | 1.00 | 1.00 | 1.00 | 1.00 |
| EAGLE | 1.86 | 1.69 | 1.69 | 1.44 | 1.19 |
| VSD | 2.13 | 2.03 | 1.88 | 1.61 | **1.41** |
| PARD | **3.63** | **3.16** | **2.59** | **1.90** | 1.33 |

## E  PERFORMANCE ON DEEPSEEK QWEN SERIES

Table 6 reports the acceleration effects of PARD on the DeepSeek-Qwen series, where the evaluation benchmarks consist of mathematics and code tasks.

Table 6: Performance comparison of different methods on the DeepSeek Qwen series.

| Target | Method | HumanEval | | GSM8K | | Math500 | | Average | |
|---|---|---|---|---|---|---|---|---|---|
| | | TPS | SpeedUp | TPS | SpeedUp | TPS | SpeedUp | TPS | SpeedUp |
| DS 7B | AR | 75.87 | 1.00 | 75.96 | 1.00 | 75.92 | 1.00 | 75.92 | 1.00 |
| | VSD | 97.90 | 1.29 | 129.20 | 1.70 | 122.51 | 1.61 | 116.54 | 1.54 |
| | PARD | **162.39** | **2.14** | **204.62** | **2.69** | **205.23** | **2.70** | **190.75** | **2.51** |
| DS 14B | AR | 40.74 | 1.00 | 40.78 | 1.00 | 40.72 | 1.00 | 40.75 | 1.00 |
| | VSD | 75.80 | 1.86 | 102.88 | 2.52 | 95.35 | 2.34 | 91.34 | 2.24 |
| | PARD | **103.34** | **2.54** | **130.17** | **3.19** | **133.98** | **3.29** | **122.50** | **3.01** |

## F  PERFORMANCE DIFFERENCES ACROSS DIFFERENT INFERENCE FRAMEWORKS

During our experiments, we observed that inference throughput using the Transformers library is significantly lower than that of vLLM. For methods such as vanilla SPD, this suboptimal performance of Transformers can lead to relative speedups under the same inference framework being lower than those measured on vLLM (e.g., 1.36× and 2.26×). To better reflect real-world usage scenarios, all final experiments in this paper are conducted on the vLLM framework. All experiments are conducted

under chain attention. Tree attention is an orthogonal decoding strategy and can be combined with PARD.

Inspired by GPT-Fast (Meta, 2023), we further optimized Transformers using `torch.compile` and a static key-value cache, resulting in **Transformers+**. Table tab:dif-method-comparison presents a comparison of Transformers, Transformers+, and vLLM, showing that Transformers+ is lightweight while approaching the performance of vLLM. We employ Transformers+ for development and testing throughout the study.

Table 7: Comparison of different frameworks and methods on HumanEval and GSM8K for LLaMA3.1-8B. Here, Transformers+ denotes an optimized version of Transformers.

| Target | Framework | Method | HumanEval | | GSM8K | | Average | |
|---|---|---|---|---|---|---|---|---|
| | | | TPS | Speedup | TPS | Speedup | TPS | Speedup |
| L3.1 8B | Transformers | AR | 34.36 | 1.00 | 35.90 | 1.00 | 35.13 | 1.00 |
| | | VSD | 50.52 | 1.47 | 45.24 | 1.26 | 47.88 | 1.36 |
| | | PARD | **145.47** | **4.23** | **114.65** | **3.19** | **130.06** | **3.70x** |
| | Transformers+ | AR | 76.34 | 1.00 | 76.50 | 1.00 | 76.42 | 1.00 |
| | | VSD | 185.29 | 2.43 | 160.59 | 2.10 | 172.94 | 2.26 |
| | | PARD | **336.97** | **4.41** | **275.03** | **3.60** | **306.00** | **4.00** |
| | vLLM | AR | 73.07 | 1.00 | 73.38 | 1.00 | 73.22 | 1.00 |
| | | VSD | 155.47 | 2.13 | 140.93 | 1.92 | 148.20 | 2.02 |
| | | PARD | **264.88** | **3.63** | **235.09** | **3.20** | **249.98** | **3.41** |

# G    COMPARISON WITH OTHER PARALLEL SPECULATIVE DECODING METHODS

To better align with real-world deployment scenarios, the performance comparisons in this paper are conducted within the vLLM framework. Although not included experimentally due to implementation constraints, we summarize below a conceptual comparison between PARD and other parallel speculative decoding methods in Table 8.

Table 8: Comparison of draft model design and target-model independence across different speculative decoding methods.

| Method | Draft Model Design | Target-Model Independence |
|---|---|---|
| ParallelSpec | Parallel draft head built based on the target model | Requires separate tuning for each target |
| BiTA | Prefix tuning with additional learned prefix tokens | Requires separate tuning for each target |
| PaSS | Look-ahead token embeddings trained with each target model | Requires separate tuning for each target |
| **PARD (Ours)** | Fully decoupled small draft model | One draft supports a family of target models |

# H    LLM USAGE STATEMENT

In the preparation of this paper, Large Language Models (LLMs) were used solely as an assistive tool for language polishing and minor stylistic refinement. LLMs were not involved in research ideation, experimental design, analysis of the paper.

