# OpenReview forum: "PARD: Accelerating LLM Inference with Low‑Cost PARallel Draft Model Adaptation"
_ICLR.cc/2026/Conference — ICLR 2026 Poster_

### Official Review · Reviewer_MJ3S · 2025-10-31

**Soundness:** 2
**Presentation:** 3
**Contribution:** 2
**Rating:** 4
**Confidence:** 4

**Summary:**

This paper proposes PARD, which accelerates autoregressive generation by training a target-independent draft model that generate drafts in parallel. Furthermore, the authors propose the CoD method to accelerate the training of the parallel inference model. Owing to parallel generation and target independence, PARD can quickly transfer across different model within the same series and generate drafts at a low cost, achieving greater acceleration compared to the autoregressive EAGLE.

**Strengths:**

1. The paper is well-structured, with clear and intuitive diagrams that effectively support the explanation.

2. It explores the concept of parallel draft generation, providing insightful and innovative perspectives on the topic.

**Weaknesses:**

1. The paper lacks a comparison with other similar draft acceleration methods, such as hierarchical acceleration [1,2] and draft models in diffusion models [3].

2. The experimental setup is not sufficiently detailed, with insufficient explanation of decoding strategies (e.g., chain or tree).

[1] Chen Z, Yang X, Lin J, et al. Cascade speculative drafting for even faster llm inference[J]. Advances in Neural Information Processing Systems, 2024, 37: 86226-86242.

[2] Zhao W, Huang Y, Han X, et al. Ouroboros: Speculative decoding with large model enhanced drafting[J]. arXiv preprint arXiv:2402.13720, 2024.

[3] Christopher J K, Bartoldson B R, Ben-Nun T, et al. Speculative diffusion decoding: Accelerating language generation through diffusion[J]. arXiv preprint arXiv:2408.05636, 2024.

**Questions:**

1. The experimental results in Table 1 lack reporting on the acceptance length, and the acceleration ratio of EAGLE is lower than that of other benchmark methods. It would be useful to clarify whether Tree validation is employed in this study.

2. In addition to the absence of a comparison with related methods, the related work section overlooks relevant literature on parallel generation techniques, which could offer valuable context for this study.

[1] Kou S, Hu L, He Z, et al. Cllms: Consistency large language models[C]//Forty-first International Conference on Machine Learning. 2024.

[2] Wang Y, Luo X, Wei F, et al. Make some noise: Unlocking language model parallel inference capability through noisy training[J]. arXiv preprint arXiv:2406.17404, 2024.

[3] Gao X, Xie W, Xiang Y, et al. Falcon: Faster and parallel inference of large language models through enhanced semi-autoregressive drafting and custom-designed decoding tree[C]//Proceedings of the AAAI Conference on Artificial Intelligence. 2025, 39(22): 23933-23941.

3. As indicated in Table 2, the high acceptance rate of PARD primarily stems from its consistency with existing homologous models, rather than additional training. Consequently, it is essential to conduct comparisons between PARD and other draft model acceleration approaches to better assess its relative performance.

---

> ### Author Response · Authors · 2025-11-20
> **Part 1**
>
> Dear reviewer,
>
> Thank you very much for the positive feedback regarding the paper’s structure and clarity, as well as for raising thoughtful and valuable questions. We address them in detail below.
>
> **Q1, W2:  Experimental setup, chain vs. tree attention, and EAGLE testing details.**
>
> **A**: All experiments in the paper were run under the industrial inference stack **vLLM** and use **chain attention**. We adopted this environment to measure performance under realistic, production-like serving conditions. vLLM currently provides a chain-attention implementation only; consequently the reported results (Table 1, Table 2, Figure 1, etc.) reflect chain-attention behavior. Tree attention is an orthogonal decoding strategy and can be combined with PARD, but it was not used in our reported experiments.
>
> EAGLE / EAGLE-3 testing details: All EAGLE / EAGLE-3 baselines were evaluated under the same vLLM + chain-attention setup and identical decoding settings used for PARD. The EAGLE/EAGLE-3 numbers reported in our paper are consistent with published EAGLE-3 evaluations (Eagle-3 [1], Section 4.4), when evaluated under comparable system-level settings.
>
> [1] Li, Yuhui, et al. "Eagle-3: Scaling up inference acceleration of large language models via training-time test." arXiv preprint arXiv:2503.01840 (2025).
>
> **Q2, W1: Comparison with hierarchical acceleration, diffusion-based drafters, and other parallel works.**
>
> **A**: thank you for pointing out additional lines of related work.
>
> Hierarchical acceleration approaches such as Cascade Speculative Drafting [2] and Ouroboros [3] introduce additional nested draft models (either smaller AR models or phrase-level decoders) to further accelerate drafting. Diffusion-based approaches [4] instead adopt a diffusion-style generator as the draft model to improve parallelism. Meanwhile, recent works such as CLLMs [5] and Noisy Parallel Decoding [6] employ Jacobi-style iterative refinement with noise augmentation, and Falcon [7] incorporates Coupled Sequential Glancing Distillation with a custom decoding tree to strengthen token dependency modeling.
>
> PARD adopts a mask-prediction–based strategy, which falls under a category of multi-token speculative decoding widely used in recent works such as BiTA, SPACE, ParallelSpec, PaSS, and now PARD. Our primary innovations (Lines 103–131) focus on enabling a *high-speed draft model that remains broadly applicable while keeping training cost extremely low*. In particular:
>
> * **Condition Drop Token (COD):** Mask-prediction-based method typically increases the number of input tokens by a factor of *k*, leading to substantially higher training cost. Our COD mechanism selectively drops part of these conditioning tokens without degrading inference performance, resulting in a 3× reduction in training overhead.
>
> * **Target Independence:** Most speculative decoding approaches including EAGLE, Medusa, BiTA, and SPACE—require training a separate draft model for each target model. This becomes increasingly costly and operationally inflexible when supporting a full model lineup (e.g., a LLaMA3 deployment might require eight separate draft models for LLaMA3-8B, 70B, LLaMA3.1-8B, 70B, 405B, LLaMA3.3-70B, etc.).
>   In contrast, PARD is target-independent: a single draft model can be shared across the entire model family, significantly simplifying deployment and maintenance.
>
> We selected EAGLE-3 as our main baseline because it is a leading, widely adopted industrially-relevant speculative method and provides a fair and practical comparison under vLLM. Our experiments show that PARD delivers higher net acceleration than EAGLE-3 in the same real-system setting, demonstrating the method’s effectiveness.
>
> We will expand the Related Work section to include the cited works.
>
> [2] Chen Z, Yang X, Lin J, et al. Cascade speculative drafting for even faster llm inference[J]. Advances in Neural Information Processing Systems, 2024, 37: 86226-86242.
>
> [3] Zhao W, Huang Y, Han X, et al. Ouroboros: Speculative decoding with large model enhanced drafting[J]. arXiv preprint arXiv:2402.13720, 2024.
>
> [4] Christopher J K, Bartoldson B R, Ben-Nun T, et al. Speculative diffusion decoding: Accelerating language generation through diffusion[J]. arXiv preprint arXiv:2408.05636, 2024.
>
> [5] Kou S, Hu L, He Z, et al. Cllms: Consistency large language models[C]//Forty-first International Conference on Machine Learning. 2024.
>
> [6] Wang Y, Luo X, Wei F, et al. Make some noise: Unlocking language model parallel inference capability through noisy training[J]. arXiv preprint arXiv:2406.17404, 2024.
>
> [7] Gao X, Xie W, Xiang Y, et al. Falcon: Faster and parallel inference of large language models through enhanced semi-autoregressive drafting and custom-designed decoding tree[C]//Proceedings of the AAAI Conference on Artificial Intelligence. 2025, 39(22): 23933-23941.

---

> > ### Comment · Reviewer_MJ3S · 2025-11-27
> >
> > Thank you for your response. My primary concerns remain centered on the following two points:
> >
> > 1. **Comparison with Similar Methods**
> >
> > As noted previously, the current experimental setup employs an overly limited baseline, utilizing only the Vanilla Speculative Decoding (VSD) method.
> > Given that the proposed PARD as a method for accelerating draft model decoding in a parallel manner, a comparison with other similar approaches is essential.
> > Based on the current results, I cannot determine whether the observed performance improvement originates from the proposed method itself or from a better-aligned draft model within the same model family.
> >
> > 2. **Fairness of EAGLE Results under Chain Verification**
> >
> > EAGLE's superior performance over Medusa is observed, particularly when utilizing Tree Verification.
> > This advantage is attributable to EAGLE's autoregressive nature, which allows a significantly larger search space via the tree structure.
> > In contrast, parallel decoding methods are limited to a Cartesian product approach.
> > This consideration is equally relevant to the method proposed in this manuscript (PARD).
> > I highly doubt that PARD will maintain its current advantage under a Tree Verification scheme.
> >
> > As noted above, I remain concerned about whether PARD demonstrates a fundamental performance improvement compared to similar approaches (e.g., ParallelSpec). Consequently, I intend to maintain my current score.

---

> > > ### Author Response · Authors · 2025-11-27
> > >
> > > Dear reviewer,
> > >
> > > Thank you for the thoughtful and constructive feedback. I address your concerns as follows.
> > >
> > > **Q1: Choice of Baselines and Comparison with Similar Methods**
> > >
> > > **A:** Our manuscript does not compare only against Vanilla Speculative Decoding (VSD), also compares PARD against **Eagle/Eagle3**, as shown in Figure 2 and Table 1. Since **Eagle3 is widely regarded as an industry-standard benchmark, demonstrating that PARD achieves 1.15× speedup over Eagle3 provides strong evidence of the effectiveness of our approach**.
> > >
> > > A core strength of PARD lies in its target-independent design. **Target-independence is an extremely valuable property** because it drastically reduces practical deployment and maintenance costs. Consider the LLaMA3 family: L3-8B, L3-70B, L3.1-405B, L3.3-70B, and several intermediate versions—eight models in total. Under a target-dependent method like EAGLE:
> > >
> > > * A pretrained draft model may not exist for each target model.
> > > * Training a new draft model from scratch can be prohibitively expensive for most users.
> > > * If the target model is later fine-tuned via SFT or RL, the hidden states used by the EAGLE head change, requiring draft retraining.
> > >
> > > By contrast, **PARD’s fully decoupled draft model avoids these issues altogether**. A single draft model can accelerate the entire model family—even after further fine-tuning—dramatically reducing operational costs.
> > >
> > > Comparison with Other Mask-Prediction-Based Parallel Speculative Decoding Methods: To better approximate real-world inference scenarios, all experiments in our paper are conducted under the high-performance inference framework vLLM. Although ParallelSpec, BiTA, and PaSS are not integrated into our evaluation, we summarize below the conceptual comparison between PARD and these parallel speculative decoding methods:
> > >
> > > | Method | Draft Model Design | Target-Model Independence |
> > > | - | - | - |
> > > | **ParallelSpec** | Parallel draft head built directly on the target model | ❌ Requires separate tuning for each target |
> > > | **BiTA** | Prefix tuning using learned prefix tokens | ❌ Requires separate tuning for each target |
> > > | **PaSS** | Look-ahead token embeddings trained per target model   | ❌ Requires separate tuning for each target |
> > > | **PARD (Ours)**  | Fully decoupled small draft model | **✔ One draft supports an entire family of target models** |
> > >
> > > ParallelSpec modifies the draft head of each target model; BiTA adds trainable prefix tokens; PaSS learns target-specific look-ahead embeddings.
> > > All of them remain **target-dependent**.
> > >
> > > PARD is fundamentally different: **It fully decouples the draft model from the target model**, enabling broad applicability and dramatically lowering adaptation cost—one of the key motivations behind our work.
> > >
> > > Regarding whether the performance gains come from same-family alignment or from PARD itself: these two factors work synergistically. As shown in Figure 1, homologous models exhibit high acceptance rates. Combined with PARD’s architecture, this allows the draft model to achieve both **high acceptance** and **low latency**, which is a central contribution of our method.
> > >
> > > **Q2: Fairness of EAGLE Results under Chain/Tree Verification**
> > >
> > > **A**: We agree that tree attention can enhance decoding efficiency, and Eagle2’s dynamic tree selection further improves performance. **Importantly, PARD is fully compatible with Eagle2-style tree attention and can benefit from the same mechanisms**. To better reflect real-world usage scenarios, all evaluations in our paper are conducted using the vLLM framework. Since vLLM does not yet support tree attention, all experiments are performed under chain attention.
> > >
> > > Below is an example illustrating how tree attention can be applied within PARD:
> > >
> > > Consider input "$2 + 1$" with $K_{\text{infer}} = 2$.
> > > The draft model input token is:
> > > $[2, +, 1, \text{mask}]$.
> > >
> > > Suppose the token probability distributions (formatted as `[token, probability]`) after one forward pass are:
> > >
> > > * First token: $[=, 0.5], [+, 0.3], [1, 0.1], \dots$
> > > * Second token: $[3, 0.6], [1, 0.2], [+, 0.1], \dots$
> > >
> > > As in Eagle2, we compute the probability of each token path, e.g.:
> > >
> > > $$
> > > P(=, 3 \mid 1, +, 2, \text{mask}) = 0.5 \times 0.6 = 0.3.
> > > $$
> > >
> > > $$
> > > P(+, 1 \mid 1, +, 2, \text{mask}) = 0.3 \times 0.2 = 0.06.
> > > $$
> > >
> > > If a three-node draft tree is desired, the tree indices are
> > > ${[0], [1], [0,0]}$.
> > >
> > > For a six-node tree, the indices become
> > > ${[0], [1], [2], [0,0], [1,0], [0,1]}$.
> > >
> > > After constructing the draft tree, verification proceeds exactly as in the EAGLE2 workflow. Thus, PARD can fully leverage dynamic tree-based verification rather than being limited to Cartesian-product expansion, and its performance is not tied to chain-only verification.
> > >
> > > ---
> > >
> > > Thank you again for your valuable comments. We hope our clarifications address your concerns. We believe the properties above demonstrate that PARD offers both conceptual and practical advantages over existing similar methods.

---

> > > > ### Comment · Reviewer_MJ3S · 2025-11-27
> > > >
> > > > I appreciate your timely feedback. I would like to further elaborate on my perspective regarding these two issues:
> > > >
> > > > 1. **Comparison with Additional Baselines**
> > > >
> > > > While I acknowledge the importance of decoupling the target and draft models, several comparable methods operating under identical settings are currently overlooked.
> > > > For instance, Ouroboros utilizes Lookahead decoding to accelerate the draft model in a training-free manner.
> > > > It is essential to compare the proposed method against such approaches to demonstrate that performance gains stem from the efficacy of parallel generation, rather than merely from the superior alignment of Qwen.
> > > > I think Eagle3 would achieve superior performance while maintaining a training budget comparable to that of the independent draft models.
> > > >
> > > > 2. **Feasibility of Tree Verification**
> > > >
> > > > To clarify, I do not deny the compatibility between parallel decoding and tree verification.
> > > > However, current parallel decoding methods—as you indicate—employ a Cartesian product approach for tree construction. The key point I wish to emphasize is that this yields a significantly smaller search space compared to the autoregressive Eagle method.
> > > >
> > > > In your example, for the autoregressive draft model, the distributions for the second token are independent. Specifically, $P(3 \mid 1, +, 2, =)$ and $P(1 \mid 1, +, 2, +)$ are sampled respectively from the outcomes of $P(= \mid 1, +, 2)$ and $P(+ \mid 1, +, 2)$, rather than remaining within the same distribution as is currently the case.

---

> ### Author Response · Authors · 2025-11-20
> **Part 2**
>
> **Q3: Homologous drafts models, and comparison to other speculative decoding approaches**
>
> **A**: As shown in Figure 1 of the paper, VSD demonstrates substantially higher acceptance rates compared to EAGLE, which directly motivates our work:
>
> If we can reduce the draft cost of an AR-based VSD model, the resulting system can achieve significantly higher overall acceleration.
>
> PARD contributes exactly in this direction by low-cost conversion of an autoregressive draft model into a parallel draft model, preserving VSD’s high acceptance rate while greatly reducing draft latency.
>
> For the comparison with other speculative decoding approaches. In the paper We compare PARD against vanilla speculative decoding and mainstream EAGLE / EAGLE-3 in the paper. Concretely, in our high-performance vLLM experiments (Table 1):
>
> * On LLaMA3.1-8B, PARD yields higher wall-clock acceleration than EAGLE-3 (PARD is ~1.15× faster than EAGLE-3 on this workload). Eagle3 is a state-of-the-art method in the industry, and comparing against it demonstrates the effectiveness of our approach.
> * PARD simultaneously provides **low training/adaptation cost**, **low draft latency**, **high draft acceptance**, and **target independence**, as summarized in Figure 1(c).
>
> ---
>
> Thank you very much for your careful review and valuable feedback. We hope our clarifications help, and we would greatly appreciate your consideration in the final evaluation.

---

> ### Author Response · Authors · 2025-12-03
>
> Thank you very much for your detailed and insightful feedback. We appreciate the opportunity to clarify our methodology. To address your comments more thoroughly, we address your concerns point by point and respond to each point individually. In addition, we have conducted supplementary experiments to further support our claims.
>
> ---
>
> **Q: Choice of Baselines & Tree Attention**
>
> **A:** To approximate real-world inference scenarios, all experiments in our paper are conducted under the high-performance **vLLM** framework. Some additional baselines (e.g., diffusion-based methods, Jacobi-style methods) and tree-attention–based verification are not yet supported in vLLM; this is the main reason they are not included in our experimental evaluation.
>
> We agree that parallel-draft and autoregressive-draft methods expand their search trees differently. We will include in the appendix a brief explanation of how PARD can incorporate tree attention.
>
> We selected **EAGLE/EAGLE-3** as baselines because vLLM supports them natively, and because they are widely considered **strong industry-standard benchmarks**. Demonstrating that **PARD achieves a 1.15× speedup over EAGLE-3** under identical conditions provides strong evidence of the effectiveness of our approach.
>
> ---
>
> **Q: Training Budget**
>
> **A:** As analyzed in **Appendix A**, using LLaMA3.3-70B as the target model, **PARD achieves 7× higher training efficiency than EAGLE and 10× higher than EAGLE-3**.
>
> Beyond training efficiency, PARD offers **target independence**, meaning that a single draft model can accelerate an entire series of target models (e.g., LLaMA3-8B, 70B, 405B). EAGLE-style methods, by contrast, require training one draft model per target, which substantially increases adaptation and maintenance cost.
>
> This efficiency difference explains why training a PARD draft model is substantially less costly in practice.
>
> ---
>
> **Q: Is the Speedup Mainly Due to Model Homogeneity?**
>
> **A: **This is an excellent question. It is true that using a same-family model—for example, LLaMA3.2-1B as the draft for LLaMA3.1-8B yields a high acceptance rate (as shown in Figure 1).
>
> **However, PARD’s acceleration does not rely on homogeneity.** To demonstrate this, we conducted a controlled experiment using LLaMA3.1-8B as the target model on HumanEval, keeping all training and inference hyperparameters identical. The experiment compares:
>
> 1. **Baseline**: No speculative decoding
> 2. **PARD (same-family draft)**: LLaMA3.2-1B → training to PARD_L3.2_1B
> 3. **PARD (random init)**: Randomly initialized draft model → training to PARD_L3.2_1B_random_init
> 4. **PARD (cross-family draft)**: Qwen2.5-0.5B adapted via vocabulary replacement → training to PARD_Q2.5_0.5B_vocab_modify
>
> Because LLaMA and Qwen use different vocabularies, we first adapted Qwen2.5-0.5B by replacing its vocabulary with LLaMA’s. We found that the two vocabularies share **85% token overlap (109,566 / 128,256)**. The replacement rule is:
>
> * If a token exists in both vocabularies, copy its original Qwen embedding.
> * If it exists only in LLaMA’s vocabulary, initialize the embedding with a normal distribution.
>
> Using this adapted model (“Q2.5_0.5B_modify”), we trained it under the PARD paradigm.
>
> | Index | Target      | Draft                 | Draft Init Method                   | TPS    | Speedup |
> | ----- | ----------- | --------------------- | ----------------------------------- | ------ | ------- |
> | 1     | LLaMA3.1-8B | —                     | —                                   | 72.70  | 1.00    |
> | 2     | LLaMA3.1-8B | PARD_L3.2_1B            | Init from LLaMA3-1B                 | 254.00 | 3.49    |
> | 3     | LLaMA3.1-8B | PARD_L3.2_1B_random_init      | Random Init                         | 94.26  | 1.30    |
> | 4     | LLaMA3.1-8B | PARD_Q2.5_0.5B_vocab_modify | Init from Qwen w/ vocab replacement | 251.47 | 3.46    |
>
> The cross-family draft model (Index 4) achieves a **3.46× speedup**, nearly identical to the same-family draft (3.49×), confirming that PARD’s benefits generalize beyond model-family homogeneity. From these results, we conclude that:
>
> * **Homogeneity is not the primary factor driving PARD’s speedup**: even a small **Qwen** draft model can deliver comparable acceleration when applied to the **LLaMA3** family after PARD training.
> * The final speedup is strongly correlated with the **intrinsic quality/capability of the draft model initialization**: small drafts from both the **Qwen** and **LLaMA3** families yield strong acceleration on **LLaMA3.1-8B**, whereas a randomly initialized draft performs substantially worse.
> * The key contributors are **(1) baseline draft model quality + (2) the PARD training paradigm**, rather than shared model family.
>
> ---
>
> We sincerely appreciate your constructive comments. Your feedback has helped us strengthen the clarity and rigor of the work, and we believe the additional analyses and experimental results address the key concerns raised.

---

### Official Review · Reviewer_GWvT · 2025-10-31

**Soundness:** 2
**Presentation:** 2
**Contribution:** 2
**Rating:** 2
**Confidence:** 4

**Summary:**

The paper proposes PARD (PARallel Draft), a novel method for speculative decoding (SD) designed to accelerate Large Language Model (LLM) inference while featuring target-independence and parallel token prediction. PARD addresses the high adaptation cost of target-dependent methods like EAGLE by allowing a single, low-cost draft model to be applied across an entire family of target models. The core mechanism involves fine-tuning a small auto-regressive (AR) model into a parallel draft model, utilizing the Mask-Predict approach with special mask tokens ($m_k$) to predict multiple candidate tokens in a single forward pass. This significantly reduces the draft model's latency from $K \times T_D$ to $T_D$.
To reduce the training cost for this parallel adaptation, the authors introduce the Conditional Drop-token (COD) mechanism. COD enables sparse sampling of training data while ensuring the integrity of prefix key-value states, boosting training efficiency by up to $3\times$ compared to traditional masked prediction training. Experiments on the industrial vLLM framework show that PARD achieves a state-of-the-art speedup on LLaMA3.1-8B.

**Strengths:**

1. This paper proposes PARD, which allows a single same-family draft model to accelerate an entire family of target models via multi-token prediction with look-head tokens.

2. This paper integrates a token-dropping scheme to reduce the draft model’s training cost.

**Weaknesses:**

- The paper frames advanced SD methods as "target-dependent" and PARD as "target-independent", yet PARD crucially assumes the availability of a same-family small model—often unrealistic for fine-tuned production LLMs. In those settings, self-speculative decoding from the target model itself (e.g., LayerSkip/ACL’24, Self-SD/ACL’24, SWIFT/ICLR’25) seems strictly more applicable. The paper should explicitly discuss this constraint, compare against self-SD scheme, and clarify when PARD is preferable.

- Much of the method resembles transferring look-ahead token prediction to a small model and then boosting with fine-tuning. Given that small-model training is a one-off offline cost, the contribution of the token dropping mechanism reads like a patch rather than a core innovation.

- The concept of parallel decoding is not novel. For instance, [1] has proposed a similar idea based on a parallel-decoded drafter and a target model as a verifier for speculative decoding. Additionally, Medusa, Multi-Token Prediction (MTP) are existing parallel decoded speculation approaches. Compared with the existing works, this paper mainly differentiates in its draft model architecture design that leverages the [MASK] tokens as a placeholder to predict future tokens within one model forward pass. However, this design is similar to the approach proposed in [2] [3].

- PARD comes from converting the sequential prediction to a parallel prediction, reducing the time cost of the draft stage from $$K \times T_D$$ to $T_D$. However, the unresolved issue is whether the $T_D^$ of the new draft model will change, and whether it is possible that $K \times T_D$ is less than $T_D^$ when the draft model is too large.

- The paper claims that its draft model is target-independent and can accelerate an entire family of target models. However, for target model families with a large range of parameter quantities, using the same draft model (lacking the adjustment of the draft model to fit the size of the target model) may lead to the problems mentioned in the last weakness. That is, universal draft model may perform slower in parallel decoding than the ideal draft model even if in sequential decoding. The evidence in the paper is insufficient to reveal or fix this issue.

- The Shared Mask Token ID Strategy in ablation study needs concrete details: how mask tokens are inserted across variants, which IDs are shared.

- Presentation issues:
   * Fig. 4 (especially subfigures (b) and (c) on training-time attention masks) is hard to parse. In addition, for Fig. 4(a), clarify the meaning of green/red. If green denotes permitted attention, the causal mask for query at position t should be upper-triangular over keys/values (e.g., "Q(story)" must attend to its history; the last column should be green).
   * Equations (7) and (8) have an indexing bug: substituting k=0 (or k=1) yields m_{-1}; define base cases and boundary conditions to prevent invalid indices.
   * The core idea of COD ("earlier-subtask tokens are more critical; later ones can be dropped") lacks justification—why are later tokens less important? Is this theoretically motivated or empirically observed (e.g., uncertainty, acceptance rates)?

**Questions:**

Please refer to Weaknesses.

---

> ### Author Response · Authors · 2025-11-20
> **Part 1**
>
> Dear reviewer,
>
> Thank you very much for your detailed feedback and for raising many valuable and insightful concerns. We address each of your questions below.
>
>
> **W1: Clarify the advantages of being “target-Independent” and compare against self-speculative methods (e.g., LayerSkip/ACL’24, Self-SD/ACL’24, SWIFT/ICLR’25).**
>
> **A**: This is an excellent question. We first clarify our definition of *target-independence*:
>
> > A draft model is *target-independent* if, once trained, it can be directly applied to accelerate multiple models within a family without any additional fine-tuning or retraining.
>
> For example, a single PARD-LLaMA3-1B model can accelerate the entire LLaMA3 series. In contrast, methods such as EAGLE and Medusa rely on target-model hidden states as input and therefore require training a separate draft model per target model, making them inherently *target-dependent*.
>
> **Target-independence is an extremely valuable property** because it dramatically reduces the cost of practical deployment. Take the LLaMA3 family as an example: the series includes L3-8B, L3-70B, L3.1-405B, and up to L3.3-70B, totaling eight models. If a user wishes to accelerate any one of these models using speculative decoding, target-dependent methods like EAGLE may not provide an available pretrained draft model, and training one from scratch is often prohibitively expensive for most users. Moreover, if the target model is further fine-tuned (e.g., via SFT or RL), the hidden states used as inputs to the EAGLE head will change, requiring the draft model to be retrained again. In contrast, PARD’s target-independent design eliminates this issue entirely: a single draft model can be used to accelerate the entire model family, even if the target models are fine-tuned afterward, significantly reducing operational and maintenance costs.
>
> **Relationship to self-speculative decoding**: Self-speculative decoding is an important research direction, but whether a method is “self-speculative” and whether it is “target-independent” are orthogonal. We clarify with two representative examples:
>
> * **LayerSkip (ACL’24):**
>   Requires training models with layer dropout and later using subsets of layers as drafts. Each target model must be individually retrained to enable self-speculation, so it is **not target-independent**.
>
> * **SWIFT (ICLR’25):**
>   Requires Bayesian optimization to determine which layers to retain for each target model. Again, this process is repeated per target model, making it **not target-independent**. SWIFT is effective but the acceleration gain is modest (typically 1.3×–1.6×).
>
> Self-speculative methods are valuable, and we will expand the Related Works section to discuss them. PARD targets a different use case: enabling a single universal draft model for an entire model family with no per-target adaptation cost, which is crucial for industrial deployments where one service must support multiple model variants simultaneously.
>
>
> **W2: The Importance of the Condition-Drop Token (COD) Mechanism.**
>
> **A**: Our method is designed to achieve high acceleration while minimizing the adaptation cost of the draft model. To this end, PARD is made **target-independent**, meaning that models within the same family require only a single adaptation. In addition, we introduce the **Condition-Drop Token (COD)** mechanism, which further reduces this one-time adaptation cost by **3×**. It is worth noting that even without COD, the draft model can still reach the same inference speedup, but COD enables us to achieve this performance at significantly lower training cost.
>
> In implementation, COD also has its own ingenuity. As detailed in Appendix B, naively dropping condition tokens at random would result in incomplete preceding KV caches, causing a mismatch between training and inference. Our approach avoids this issue by ensuring that the KV cache before the condition remains complete, and only then performs token dropping. This allows PARD to be trained with acceleration benefits without introducing performance inconsistency.
>
> Therefore, COD is an essential mechanism that makes PARD practical and cost-efficient for real-world deployment.

---

> ### Author Response · Authors · 2025-11-20
> **Part 2**
>
> **W3: Explain the relationship between PARD and parallel decoding, and how parallel decoding methods such as Medusa and Multi-Token Prediction relate to mask prediction.**
>
> **A**: Since the reviewer did not specify which prior works they were referring to, I will answer based on my understanding.
>
> I address two aspects separately: **(1) the relationship between PARD and parallel decoding**, and **(2) how parallel decoding methods such as Medusa and multi-token prediction relate to mask prediction**.
>
> **Relationship between PARD and parallel decoding.**
> PARD adopts a mask-prediction–based parallel decoding strategy, which falls under a category of multi-token speculative decoding widely used in recent works such as BiTA, SPACE, ParallelSpec, PaSS, and now PARD. Parallel decoding itself is not our main contribution. Instead, our primary innovations (Lines 103–131) focus on enabling a *high-speed draft model that remains broadly applicable while keeping training cost extremely low*. In particular:
>
> * **Condition Drop Token (COD).**
>   Mask-prediction-based parallel decoding typically increases the number of input tokens by a factor of *k*, leading to substantially higher training cost. Our COD mechanism selectively drops part of these conditioning tokens without degrading inference performance, resulting in a **3× reduction in training overhead**.
>
> * **Target Independence.**
>   Most speculative decoding approaches—including EAGLE, Medusa, BiTA, and SPACE—require training a separate draft model for each target model. This becomes increasingly costly and operationally inflexible when supporting a full model lineup (e.g., a LLaMA3 deployment might require eight separate draft models for LLaMA3-8B, 70B, LLaMA3.1-8B, 70B, 405B, LLaMA3.3-70B, etc.). In contrast, PARD is *target-independent*: a single draft model can be shared across the entire model family, significantly simplifying deployment and maintenance.
>
> **Difference between PARD and Medusa-style parallel decoding.**
> Although both PARD and Medusa are parallel decoding, their computational properties are fundamentally different. In particular, PARD’s memory bandwidth consumption during decoding does not grow with the number of candidate tokens $k$, whereas Medusa’s does, because each Medusa head maintains its own independent parameters. For example, using LLaMA3-70B as a reference:
>
> * **PARD:**
>   Regardless of the value of (k), the memory bandwidth cost of the draft model is constant and determined solely by the 1B-parameter draft model size.
>
> * **Medusa:**
>   Each prediction head is an independent set of parameters. With 5 parallel heads, the parameter size is approximately
>   $$ 5 \times \bigl(\text{HiddenDim}^2 + \text{HiddenDim} \times \text{VocabSize}\bigr) \approx 5.6\text{B} $$
>   Since LLM decoding is **memory-bound**, this means Medusa’s memory bandwidth cost scales *linearly* with $k$, while PARD’s cost stays fixed.
>
> In summary, PARD leverages the advantages of Mask-prediction-based parallel decoding while introducing new mechanisms that make the draft model cheaper to train, easier to deploy, and more efficient to run in large real-world systems.
>
>
> **W4: In the modeling, does $T_D$ get affected by introducing a “new draft model”?**
>
> **A**: The execution of $T_D$ is identical in nature to $T_T$, it performs masked multi-token prediction in a single forward pass. The assumption that multiple tokens can be produced in parallel without increasing latency is standard in speculative decoding literature, e.g.:
>
> > *“Fast Inference from Transformers via Speculative Decoding”, Section 3.3.*
>
> Therefore, introducing PARD does not fundamentally change the latency model for $T_D$.
>
> **W5: When using the same model to accelerate a series of target models, what impact does this have on target models of different sizes?**
>
> **A**: In Table 1 of the paper, we report the speedup achieved by using the same draft model to accelerate target models of different sizes. Specifically, L3.1‑8B and L3.3‑70B achieve speedups of 3.00× and 3.49×, respectively. In the table below, we re-evaluated these two models with respect to their accepted lengths, which are very close. This further demonstrates that a single draft model can effectively accelerate different target models.
>
> | Target Model | Benchmark | Speed-up | Accept Length |
> | ------------ | --------- | -------- | ------------- |
> | LLaMA3.1-8B  | HumanEval | 3.72×    | 5.72          |
> | LLaMA3.3-70B | HumanEval | 4.15×    | 5.83          |
>
> The similar acceptance lengths indicate that the same draft model generalizes well to different model sizes, empirically supporting the feasibility of universal drafting for model families.

---

> ### Author Response · Authors · 2025-11-20
> **Part 3**
>
> **W6: The Shared Mask Token ID strategy in the ablation study needs concrete details: how mask tokens are inserted across variants and which IDs are shared.**
>
> **A**: We appreciate the request for clarity. In practice, we simply reuse existing special token IDs in the draft model vocabulary:
>
> | Draft Model  | Shared Token ID | Unshared token IDs  |
> | ------------ | --------------- | ------------------- |
> | PARD-Qwen2.5 | [151665]        | [151665, 151666, …] |
> | PARD-LLaMA3  | [128020]        | [128020, 128021, …] |
>
> We will add these configurations to the appendix for completeness.
>
>
> **W7: Presentation issues in Fig. 4 and indexing in Eqs. (7)(8).**
>
> **A**: Thank you very much for your careful observation. In Fig. 4, the KV and Q were mistakenly swapped, and we will correct this in the final version.
>
> Regarding Equations (7) and (8), we originally wrote them following the code implementation, which may appear unclear when reading. We will revise them for clarity in the final version. For example, in Equation (7) from the paper, when $k = 0$, the first term on the right-hand side degenerates to a form without any $m_i$ as input.
> $$
> P(x_n \mid x_0, \dots, x_{n-1}, m_0, \dots, m_{-1}) =
> P(x_n \mid x_0, \dots, x_{n-1})
> $$
>
>
> **W8: Why are earlier-subtask tokens more critical, while later ones can be dropped?**
>
>
> **A**: This is an excellent question, and we would like to explain the motivation behind this design. In speculative decoding, the total acceptance length can be defined as:
>
> $$
> l = 1 + p_1 + p_2 + \dots + p_K = 1 + \alpha + \alpha^2 + \dots + \alpha^K,
> $$
>
> where $l$ is the total acceptance length, $K$ is the number of candidate tokens generated by the draft model, $p_i$ is the probability that the (i)-th candidate token is accepted, and $\alpha$ is the acceptance rate. Here, we assume that the acceptance rate is identical across different positions, which holds for vanilla SD.
>
> We can interpret $p_i$ as the contribution of the (i)-th candidate to the overall acceptance length. From the formula above, the probability of acceptance naturally decreases exponentially for later positions—the further the token, the lower its contribution.
>
> In mask prediction, different training tasks correspond to ensuring the acceptance of the (i)-th candidate, i.e., $p_i$. It is therefore natural to allocate more training emphasis on earlier, more critical tokens, while reducing emphasis on later, less important tokens. Based on this analysis, the COD mechanism is designed according to the principle: **earlier-subtask tokens are more critical, while later ones can be dropped**.
>
> ---
>
> Thank you again for your thoughtful and detailed reviews. We believe the clarifications above address your concerns, and we hope our responses improve your evaluation of the work.

---

> ### Comment · Reviewer_GWvT · 2025-11-26
>
> Thanks to the authors for the details responses. The rebuttal has partially addressed my concerns. I would raise the score accordingly.

---

### Official Review · Reviewer_Gfez · 2025-10-31

**Soundness:** 3
**Presentation:** 3
**Contribution:** 3
**Rating:** 6
**Confidence:** 4

**Summary:**

This paper proposes PARD (Parallelized Accelerated Reasoning Decoding), a novel speculative decoding method featuring target-independence and parallel token prediction. PARD is highly generalizable: its target-independent design allows a single draft model to accelerate an entire family of target models. Through mask tokens, PARD substantially accelerates inference by predicting multiple future tokens within a single forward pass of the draft phase. And they propose a COnditional Drop-token (COD) strategy for the effient training of PARD. Leveraging the integrity of prefix key-value states, COD enables low-cost adaptation of autoregressive draft models into parallel ones, boosting training efficiency by up to 3× while maintaining accuracy. In the end, the author integrates PARD into the high-performance inference framework vLLM.

**Strengths:**

1. This paper makes a pivotal conceptual shift by introducing a target-independent speculative decoding framework. Unlike dominant approaches like EAGLE and Medusa that train a draft model tightly coupled to a specific target model, PARD enables a single draft model to accelerate an entire family of target models.
2. The work innovates by integrating parallel token prediction into speculative decoding via masked language modeling.A key novelty is the Conditional Drop-token (COD) mechanism, which leverages the integrity of prefix key-value states to dramatically reduce training overhead. This allows efficient adaptation of autoregressive models into parallel draft models. The use of a shared mask token ID to enable extrapolation beyond the trained prediction length is another clever and inventive design choice that enhances practicality.
3. The integration with the industrial-grade vLLM framework underscores its immediate practical significance, providing a scalable and efficient solution that directly addresses real-world inference bottlenecks.

**Weaknesses:**

Limited Analysis of the Parallel Drafting Mechanism's Limitations: The paper celebrates the efficiency of parallel drafting but fails to critically analyze its inherent constraints. Generating K tokens in one forward pass necessarily uses less contextual information (relying on masks) than an autoregressive model, which may hurt prediction accuracy for long-range dependencies or complex reasoning steps. This likely explains the performance drop on certain tasks (e.g., PARD vs. EAGLE-3 on SpecBench for LLaMA3.3-70B). The work would be strengthened by a dedicated analysis—for instance, measuring how acceptance rate decays with K on different task types—to delineate the practical boundaries of its parallel approach.

**Questions:**

N/A

---

> ### Author Response · Authors · 2025-11-20
>
> Dear reviewer,
>
> Thank you very much for your constructive feedback and for highlighting this important question.
>
> **W1: How does the parallel drafting mask mechanism affect accuracy and acceptance rate as the prediction length K increases?**
>
> **A**: This is an excellent question. While masked parallel prediction significantly improves decoding efficiency, it inevitably has an impact on acceptance rate as the prediction length increases. In the paper (Table 2), we report acceptance rates for EAGLE/EAGLE-3 and PARD. For greater clarity, we additionally include the results from a vanilla autoregressive Speculative decoding draft (VSD, without masking) and organize the comparison below, where $k\text{-}\alpha$ denotes the average acceptance rate when the draft length is $k$:
>
> | Method          | HumanEval 1-α | HumanEval 4-α | GSM8K 1-α | GSM8K 4-α |
> | --------------- | ------------- | ------------- | --------- | --------- |
> | EAGLE           | 0.83          | 0.72          | 0.79      | 0.66      |
> | EAGLE-3         | 0.87          | 0.85          | 0.82      | 0.79      |
> | VSD             | 0.94          | 0.94          | 0.89      | 0.89      |
> | **PARD**        | 0.93          | 0.90          | 0.88      | 0.85      |
>
> **Interpretation and analysis**: VSD achieves the highest acceptance rate among all methods, which is precisely one of the motivations behind PARD: leveraging VSD’s strong acceptance properties while accelerating inference via parallel masked prediction. Because VSD generates tokens autoregressively, each draft position yields identical acceptance behavior, making $k\text{-}\alpha$ effectively constant.
>
> When VSD is converted into a PARD model:
>
> * **For $K=1$**, PARD and VSD have nearly identical acceptance rates. This is because substituting (K=1) into Eq. (6) and Eq. (7) in the paper makes the prediction objectives equivalent:
>   $$
>   P(x_n | x_0,\dots,x_{n-1}; \theta_{\text{AR draft}})
>   \quad \text{and} \quad
>   P(x_n | x_0,\dots,x_{n-1}; \theta_{\text{PARD}})
>   $$
>
>   are modeled the same way.
>
> * **For $K>1$**, the formulation becomes:
>   $$
>   P(x_{n + K -1} | x_0, \dots, x_{n + K - 2};  \theta_{\text{AR draft}})
>   \quad \text{and} \quad
>   P(x_{n+k-1} | x_0,\dots,x_{n-1}, m_0,\dots,m_{k-2}; \theta_{\text{PARD}})
>   $$
>   where placeholders $m_i$ must stand in for future context. This makes the prediction task inherently more challenging than AR drafting without placeholders, which explains the gradual decay in acceptance rate.
>
> **Empirical outcome**: Despite this, the degradation is relatively minor:
>
> * Even at $K=4$, masked parallel decoding reduces acceptance rate by only **~4%** on average.
> * In return, it reduces the number of draft forward passes by a factor of **$1/K$**.
>
> Thus, when balancing acceptance rate and draft speed, PARD consistently provides a higher net acceleration while maintaining strong accuracy, making it a practical solution for real-world parallel speculative decoding.
>
> ---
>
> Thank you again for raising this insightful point. We hope this clarification addresses your concerns and helps improve your evaluation of our work.

---

> ### Comment · Reviewer_Gfez · 2025-11-26
>
> Thanks for your detailed follow-up explanations. I’ll maintain my original score, which still reflects a positive evaluation of your work.

---

### Official Review · Reviewer_6iwh · 2025-10-31

**Soundness:** 3
**Presentation:** 3
**Contribution:** 2
**Rating:** 6
**Confidence:** 3

**Summary:**

The paper introduces PARD, a speculative decoding framework that accelerates LLM inference through a target-independent, parallel draft model. It combines mask-based prediction and a novel Conditional Drop-token (COD) training strategy to enhance both training and inference efficiency. PARD achieves up to 3.67× speedup on LLaMA3.1-8B and outperforms the current state-of-the-art EAGLE-3 model in both speed and generalizability.

**Strengths:**

1. Unlike methods like EAGLE, PARD can accelerate an entire family of target models with a single draft model, significantly reducing adaptation and deployment costs.
2. Moreover, the use of masked tokens allows PARD to predict multiple tokens in a single pass, improving inference speed without sacrificing quality.
3. Extensive benchmarks on HumanEval, GSM8K, and SpecBench across multiple LLMs demonstrate consistent speedups and high acceptance rates compared to EAGLE and vanilla SD.

**Weaknesses:**

The paper introduces PARD as a parallel and target-independent speculative decoding method. However, the experimental section lacks direct comparisons with other recent parallel speculative decoding methods such as ParallelSpec, BiTA, or PaSS.

**Questions:**

Could the authors justify the omission of comparing to other recent parallel speculative decoding methods, and if possible, include some comparative results to support the performance and efficiency claims of PARD in the parallel decoding category?

---

> ### Author Response · Authors · 2025-11-20
>
> Dear reviewer,
>
> Thank you for recognizing the strengths of our work and for raising this valuable question. We address your concern below.
>
> **W1,Q1: Why does the paper not include direct comparisons with other recent parallel speculative decoding methods (e.g., ParallelSpec, BiTA, PaSS), and can the authors justify or supplement this omission?**
>
> **A**: This is an excellent question. We first clarify the rationale behind our baseline selection in the experiments, and then provide a methodological comparison between PARD and other recent parallel speculative decoding approaches.
>
> **Baseline selection and evaluation environment:** Our primary goal was to evaluate performance in realistic, production-level scenarios. Therefore, all experiments were conducted under **vLLM**, a widely used high-performance inference engine in modern LLM services. Appendix F of the paper compares vLLM and the standard *Transformers* inference environment commonly used in academic work:
>
> * For autoregressive baseline decoding, vLLM achieves nearly **2× the decoding speed** compared to Transformers.
> * For VSD-style decoding, Transformers only yields **1.36× speedup**, while vLLM achieves **2.02× speedup**.
>
> These results indicate a large gap between academic results obtained under Transformers and real-world serving performance. Since methods such as ParallelSpec, BiTA, and PaSS do not provide vLLM implementations, including them only with Transformers would create an unfair comparison and would not reflect the practical deployment environment. Therefore, we selected **EAGLE and EAGLE-3**, which currently offer the strongest real-system improvements and support high-performance inference stacks, as our main baselines.
>
> **Methodological comparison:** Although not included experimentally due to implementation constraints, we summarize below how PARD compares conceptually with other recent parallel speculative decoding methods:
>
> | Method          | Draft Model Design                                         | Target-Model Independence                          |
> | --------------- | ---------------------------------------------------------- | -------------------------------------------------- |
> | ParallelSpec    | Parallel draft head built base on the target model          | ❌ Requires separate tuning for each target           |
> | BiTA            | Prefix tuning with additional learned prefix tokens        | ❌ Requires separate tuning for each target         |
> | PaSS            | Look-ahead token embeddings trained with each target model | ❌ Requires separate tuning for each target            |
> | **PARD (Ours)** | Fully decoupled small draft model                          | **✔ One draft supports a family of target models** |
>
> ParallelSpec modifies the draft head and still requires per-model optimization. BiTA adds trainable prefix tokens, again requiring customized tuning for each target. PaSS learns look-ahead embeddings that are tied to individual target models.
>
> In contrast PARD fully decouples the draft model from the target model. A single small draft model can accelerate an entire *family* of target models without retraining, significantly reducing adaptation and deployment cost—one of the key motivations for the method.
>
> We will include an expanded discussion of these differences in the revised version.
>
> ---
>
> Thank you again for your constructive comments. We hope our clarifications adequately address your concerns and help improve your evaluation of the work.

---

### Meta-Review · Area_Chair_nwsp · 2026-01-13

**Summary:**

This paper proposes PARD, a speculative decoding method that combines target-independence with parallel token prediction using mask-based drafting. Key strengths include: (1) a single draft model can accelerate an entire family of target models without per-target retraining, significantly reducing deployment costs; (2) the COD (Conditional Drop-token) mechanism reduces training cost by 3x while maintaining inference performance; (3) strong empirical results showing 3.67x speedup on LLaMA3.1-8B and 1.15x faster than EAGLE-3 under vLLM; (4) the method is simple and well-motivated. Weaknesses include limited comparison with other parallel speculative decoding methods (ParallelSpec, BiTA, PaSS) due to lack of vLLM implementations, and questions about whether speedups stem from same-family alignment rather than the method itself.

**Reviewer Concerns:**

Addressed: (1) The family alignment concern was convincingly addressed with a cross-family experiment showing a Qwen2.5-0.5B draft model (adapted via vocabulary replacement) achieves 3.46x speedup on LLaMA3.1-8B, nearly identical to the same-family 3.49x, demonstrating PARD's gains are not tied to model homogeneity. (2) Acceptance rate degradation as K increases was analyzed, showing only ~4% drop at K=4 while reducing draft passes by K-fold. (3) COD mechanism was justified via expected acceptance-length analysis. (4) Comparison with EAGLE/EAGLE-3 under identical vLLM settings was provided. Outstanding: Direct experimental comparison with ParallelSpec, BiTA, PaSS remains missing (though justified by lack of vLLM support), and tree attention experiments were not included (vLLM limitation).

**Reviewer Scores:**

Reviewer 6iwh (6): Would likely maintain score. Did not engage post-rebuttal but their concern about missing baselines was addressed with methodological comparison.
Reviewer Gfez (6): Maintains score. Explicitly stated they maintain their positive evaluation after the detailed acceptance rate analysis.
Reviewer GWvT (2→raised): Explicitly raised score after rebuttal, stating concerns were partially addressed. Likely raised to 4 or 6 given the thorough responses on target-independence, COD justification, and cross-family experiments.
Reviewer MJ3S (4): Would likely maintain score. Continued to express concerns about missing baselines and tree verification, though the cross-family experiment partially addressed the homogeneity concern.

---

### Decision · Program_Chairs · 2026-01-26

Accept (Poster)